# Racism as a Social Determinant of Health for Newcomers towards Disrupting the Acculturation Process

Jessica Naidu [1], Elizabeth Oddone Paolucci [1] and Tanvir Chowdhury Turin [1,2,*]

1. Department of Community Health Sciences, Cuming School of Medicine, University of Calgary, Calgary, AB T2N 4N1, Canada
2. Department of Family Medicine, Cuming School of Medicine, University of Calgary, Calgary, AB T2N 4N1, Canada
* Correspondence: turin.chowdhury@ucalgary.ca; Tel.: +1-403-210-7199

**Abstract:** Previous research has demonstrated that racism is a social determinant of health (SDOH), particularly for racialized minority newcomers residing in developed nations such as the United States, Canada, New Zealand, and European countries. This paper will focus on racism as a SDOH for racialized newcomers in these countries. Racism is defined as "an organized system of privilege and bias that systematically disadvantages a group of people perceived to belong to a specific race". Racism can be cultural, institutional, or individual. Berry's model of acculturation describes ways in which racialized newcomers respond to their post-migration experiences, resulting in one of several modes of acculturation; these are integration, assimilation, separation, and marginalization. After examining the definition and description of racism, we argue that racism impacts newcomers at the site of acculturation; specifically, the paths they choose, or are forced to take in response to their settlement experiences. We posit that these acculturation pathways are in part, strategies that refugees use to cope with post-displacement stress and trauma. To support acculturation, which is primarily dependent on reducing the effects of cultural, institutional, and individual racism, health policymakers and practitioners are urged to acknowledge racism as a SDOH and to work to reduce its impact.

**Keywords:** racism; health; social determinant of health; acculturation





## 1. Introduction

Previous research has demonstrated that racism is a social determinant of health (SDOH) that is associated with poor physical and mental health outcomes [1–4]. Moreover, racism has a unique impact on racialized minority newcomers residing in developed nations such as the United States, Canada, New Zealand, and countries in Europe [2,5]. This association has been demonstrated for many racialized populations, including indigenous communities, native-born people of colour, and newcomers of colour [6]. There is little research, however, on how racism affects acculturation [7], which is "the process by which migrants to a new culture develop relationships with the new culture and maintain their original culture [8] (p. 292)".

This paper examines racism as a SDOH for racialized newcomers in developed countries. We will define and describe racism before summarizing the existing literature on racism as a SDOH using four key literature reviews. Using Barry's typology of acculturation, we will explain how racism functions as a SDOH for racialized newcomer populations in developed countries. This analysis aims to add to a dialogue and future research on how racism impacts newcomers specifically, by interrupting the process of acculturating into a new society.

## 2. Situating the Researchers

While this is not a primary research project in which researchers interacted with participants, we must consider our writers' positionality. This article focuses on racialized minority newcomers, who are often members of marginalized communities and are often spoken about by outsiders in positions of privilege. Thus, it is critical for us as writers to consider our positionality in relation to the subjects of this article [9]. Positionality describes an individual's world view and the position they adopt about a research task and its social and political context [10–12].

One of the authors (J.N.) is a Canadian-born woman of South Asian ethnicity. She speaks English as her first language and Hindi as her second. As a Canadian-born citizen, she is in a position of privilege in terms of immigration status. She does not personally bear the fear of losing her right to residency, nor is she separated from loved ones in her birth country. She does not experience discrimination based on her English proficiency and accent because she speaks English as a first language. This researcher has proximity to newcomers through her parents and much of her extended family, thus she has some awareness of the newcomer experience. Another author is a white female (E.O.P) who was born in Canada one month after her parents emigrated from northern Italy. She is sensitive to the hardships of newcomers and the tensions of retaining cultural, ethnic, and religious values in Canadian culture. She works in academia as an applied educational psychologist, placing her in a position of privilege and influence. The last and corresponding author (T.C.T) is a Bangladesh immigrant to Canada. He is employed as an academic in Canada which places him in a position of privilege. He conducts research with racialized/ethnic-minority communities. He has observed very closely the issues of misinformed stereotyping, unconscious bias, pre-judging or prejudicing, discrimination, and racism at the structural and community levels.

The purpose of this paper is to provoke reflection and inspire discourse about racism as a SDOH and a key factor in determining newcomer acculturation. Our goal is for this work to contribute to a shift in research focusing on newcomer experiences using an anti-racist lens. We hope that such research further benefits newcomer communities by allowing us to hear their stories, validate their experiences, and work toward the goal of reducing the impact of racism on immigrant health and wellness, and ultimately to eradicate racism.

## 3. Racism

There are various definitions of racism and much debate in public discourse; therefore, it is critical that we define and describe the concept of racism adopted in this paper. We adopt Dovidio, Gaertner, and Kawakami's (2010) definition of racism, defined as "an organized system of privilege and bias that systematically disadvantages a group of people perceived to belong to a specific race [13] (p. 312)". Dovidio et al. also adhere to the sociological definition of racism as "prejudice plus power". In other words, for an event to be racist as opposed to simply prejudiced, it must further disadvantage an already marginalized racial group within a specific social context [13] (p. 316). Dovidio et al. characterize racism into three categories: (i) cultural, (ii) institutional, and (iii) individual. Each of these is described further below.

Cultural racism occurs when a racially dominant group defines cultural values for all [13]. One example of this is the declaration of Christmas day as a statutory holiday in Canada, the US, Australia, New Zealand, and European countries, with almost universal time off work and school to observe the tradition. While Christmas is celebrated by many racial groups in these countries, it is historically a "white" holiday with European roots; the adoption of Christmas by racialized minorities is directly linked to colonialism [14]. Moreover, the aforementioned countries identify as multi-cultural mosaic societies, with many racialized minority communities celebrating religious and cultural holidays other than Christmas, which are not generally recognized as statutory holidays. Recognizing Christmas as a statutory holiday whilst not recognizing other, commonly observed holidays as statutory holidays serves to establish Christmas as a norm, while "othering" holidays

such as Eid al Fitr, Diwali, and Kwanzaa [15]. The treatment of Roma communities in Europe is also reflective of cultural racism [16–18]. For example, widespread hostility toward Roma throughout European countries is often rooted in a feeling of threat toward the lifestyles and traditions of host countries. Roma culture, particularly the aspects of transience and separateness, is often perceived as incompatible with host countries. State policy and practice often promote the complete assimilation of the Roma people, thus the complete erasure of Roma culture. The Roma are seen as a threat to the host society, which creates a climate of hostility toward the Roma and allows for the social mistreatment and deprivation of Roma communities [17].

Institutional racism occurs when institutional practices have a disproportionately negative impact on racialized minorities. A practice or policy does not have to be explicitly or intentionally targeted toward racialized minorities to be termed institutional. When such practices and policies are routinely applied with impunity, they gain institutional power. The intentions of perpetrators have little to no bearing on the grievous harms inflicted by racism on its victims [13]. The Canadian practice of traffic police documenting and forwarding personal information on people of interest to detectives, colloquially known as "carding," is an example of institutional racism. Carding policies do not explicitly compel police officers to disproportionately target black, Indigenous, and people of colour (BIPOC). Nonetheless, in Toronto, Canada, BIPOC individuals are carded at a higher rate than non-BIPOC, resulting in a disproportionate rate of police violence and overrepresentation of BIPOC people in the Canadian correctional system [19]. (Henry and Tator 2016). Institutional racism can also be found in school systems, where curriculum design focuses disproportionately on ethnocentric perspectives of history, omitting the voices and experiences of racialized people. This leads to students feeling invisible and unheard within the school system, and "home education" is often more valuable than classroom instruction. Disengagement and disenfranchisement can have major consequence for school satisfaction and performance [20].

Individual racism is enacted from one person to another. It can manifest itself as prejudice, which is bias against an individual based on their perceived identity. Discrimination occurs when an individual acts upon their prejudice, resulting in unjust treatment of another individual based on their perceived identity. Racism is also frequently manifested as stereotyping, which is the misattribution of negative characteristics to an individual based on beliefs about their identifiable group [13]. For example, immigrant South Asian women in Australia who are diagnosed with gestational diabetes mellitus report experiences of stereotyping from health care providers. These experiences include assumptions that South Asian women's' health literacy and dietary behaviours are primarily responsible for their diabetes diagnoses and outcomes. During health appointments, such stereotypes cause women to feel ashamed and discouraged [21]. In the context of New Zealand, a comparable example may be discussed. Harris, Cormack, and Stanley (2019) found that racism by health professionals, which often includes race-based stereotyping and assumptions of health behaviours, leads to higher rates of unmet needs in New Zealand's racialized communities [22].

## 4. Racialization

In this paper, we refer to the concept of racialization as opposed to race. This is done to emphasize that the health outcomes we discuss are not the product of one's biology (i.e., their racial or ethnic make-up), but of the experience they have as a result of how they are perceived and treated in social settings. "Racialization is the process of manufacturing and utilizing the notion of race in any capacity [23] (p. 27)". It is the complex social and cultural process by which individuals and groups are ascribed a particular "race" and socially stratified based on that race. Racialization has been and continues to be a kind of inequitable social stratification, resulting in social and health disparities. As such, race is a social construct as opposed to a fundamental part of an individual or group. The use of race as a variable in human studies has been deemed questionable [24], and even

racist [25,26]. For example, Hunt and Megyesi (2008) conducted interviews with human genetic scientists who used race as a variable in their research. They found that the basis on which the researchers categorized individuals by race were nebulous and illogical, and that, despite claims of scientific neutrality, we live in a racist culture, which means that race is socially constructed [27]. The authors concluded that "persisting in constructing scientific arguments based on highly ambiguous variables that are clearly laden with dubious social meanings, is of deep concern [27] (p. 11)". Our paper contributes to the body of literature on racialized health disparities as opposed to racialized health outcomes by demonstrating that such health outcomes are unfair, avoidable, and socially produced, rather than an intrinsic component of an individual's biology [25,26].

## 5. Literature Review Summary

This section will summarize the findings of four key literature reviews that we have identified as critical to understanding the relationship between racism as a SDOH and newcomer health and wellness.

### 5.1. Perceived Discrimination and Health: A Meta-Analytic Review

In their meta-analytic review, Pascoe and Smart Richman (2009) focus on perceived discrimination based on race, gender, sexual orientation, and other identities rather than racism exclusively. Racism was found to be the most common type of perceived discrimination, appearing in 65% of all articles in their literature search [28]. The authors define discrimination as "a behavioural manifestation of a negative attitude, judgment, or unfair treatment toward members of a [28] (p. 3)", which is consistent with the description presented above by Dovidio, Gaertner, and Kawakami (2010). A total of 134 articles supported their hypothesis that perceived discrimination is associated with poor physical and mental health outcomes. (Pascoe and Smart Richman 2009) Discrimination, in particular, is associated with "heightened physiological stress responses, more negative psychological stress responses, increased participation in unhealthy behaviours, and decreased participation in healthy behaviours [28] (p. 20)".

### 5.2. A Systematic Review of Empirical Research on Self-Reported Racism and Health

Paradies (2006) reviewed 138 quantitative research articles and similarly found that racism is associated with poor health for oppressed racial groups. Negative mental health and mental illness were found to have the strongest associations [2]. Five key pathways between racism and health were identified; racism was related to: (1) reduced access to employment, housing and education, and/or increased exposure to risk factors such as contact with police; (2) adverse cognitive/emotional experiences and psychopathology; (3) allostatic load and concomitant patho-physiological processes; (4) reduced participation in healthy behaviours such as exercise and/or increased participation in unhealthy behaviors such as substance use; and (5) physical injury as a result of racist violence [2].

### 5.3. Racism as a Social Determinant of Health: A Systematic Review and Meta-Analysis

In their systematic review, Paradies et al. (2015), like Dovidio, Gaertner, and Kawakami (2010), define racism as "organized systems within societies that cause avoidable and unfair inequalities in power, resources, capacities, and opportunities across racial or ethnic groups [6] (p. 2)". Data from 293 studies revealed that racism was associated with lower health outcomes on all measures. Depression (37.2% of articles) was the most often reported mental health outcome, followed by self-esteem (24.3%), psychological stress (21.3%), distress (18.3%), and anxiety (14.4%). High blood pressure and hypertension were the most reported physical health outcomes, reported in 7.2% of articles [6].

### 5.4. Implicit Racial/Ethnic Bias among Health Care Professionals and Its Influence on Health Care Outcomes: A Systematic Review

Hall et al. (2015) conducted a systematic review of fifteen cross-sectional studies examining implicit bias using the Implicit Association Test [29]. They found low to moderate levels of implicit racial bias among health care professionals against racialized minorities compared with white people across all but one study. They also found that implicit bias was significantly related to four categories/themes: patient–provider interactions, treatment decisions, treatment adherence, and patient health outcomes. The most significant of these relationships was implicit bias and patient–provider interactions, with black patients perceiving poorer treatment in terms of patient centered care. Another significant relationship was and implicit bias and health outcomes, particularly psychosocial health outcomes such as social integration, depression, and life satisfaction [29].

### 5.5. Social Determinants of Health: The Impact of Racism on Early Childhood Mental Health

In their review of the impact of racism on infant and early childhood mental health and socioemotional development. Berry, Tobon, and Njoroge (2021) found that young children are particularly impacted by experiences of and indirect exposure to racism [30]. The authors conclude that" racism is particularly nefarious to young children's socioemotional development" and has long term implications for mental health into adolescence and adulthood. Racism has unique impacts on children from the perinatal period, to the infant toddler period, and into pre-school and grade school. Moreover, children are affected both directly and indirectly by racism throughout their development. For example, racialized minority children are direct victims of racism in school environments, which negatively affects their mental health. Additionally, racism affects parenting practices and maternal/caregiver mental health, which then affects which negatively affects child mental health [30].

### 5.6. The Perspectives of Health Professionals and Patients on Racism in Healthcare: A Qualitative Systematic Review

A qualitative systematic review of 23 articles, with a total of 1006 participants across the articles, looked at the perspectives of professionals and patients on racism. The authors found that healthcare providers perpetuated racism due to their unconscious (and sometimes conscious) biases toward patients. For example, providers professed less empathy toward racialized minority patients because they were less able to connect with patients of a different race than themselves, yet they often shifted blame for health disparities on minority patients behaviours as opposed to racism. The authors also found two major themes, generated through interviews with racialized minorities. These were: (1) alienation of minorities due to racial supremacism and lack of empathy, resulting in inadequate medical treatment; and (2) labelling of minority patients who were stereotyped as belonging to a lower socio-economic class and having negative behaviors. The findings of this study support the notion that experiences of racism in healthcare interactions contributes to inadequate medical service and treatment for racialized minorities, which can then lead to poor health outcomes [31].

### 5.7. Perceived Racism and Mental Health among Black American Adults: A Meta-Analytic Review

A meta-analytic review of perceived racism and mental health among Black Americans found that perceived racism is associated with poor physical and mental health outcomes [32]. The authors focused on this group because they report more incidents of racism than other racialized minorities. Pieterse et al. (2012) systematically reviewed 66 studies with total sample size of 18,140 across studies. Using a random effects model, the authors found a positive association between perceived racism and psychological distress ($r = 0.20$). As exposure to and appraised stressfulness of racist events increased, so did the likelihood of reporting mental distress. Moreover, effects for psychiatric symptoms and general distress were stronger than effects for life satisfaction and self-esteem. Thus, these

findings support the notion that racism ought to be viewed within the context of trauma in the field of mental health [32].

*5.8. The Global Refugee Crisis: Empirical Evidence and Policy Implications for Improving Public Attitudes and Facilitating Refugee Resettlement*

As part of their broader investigation into the global refugee crisis, Esses, Hamilton, and Gaucher conducted three literature reviews. The two most relevant to this paper are: (1) the determinants of public attitudes toward refugees, and (2) factors affecting refugee mental health [33].

Esses et al. use the UNHCR definition of refugees, which is "a person who is outside his or her country of nationality or habitual residence; has a well-founded fear of being persecuted because of his or her race, religion, nationality, membership of a particular social group or political opinion; and is unable or unwilling to avail him- or herself of the protection of that country, or to return there, for fear of persecution [33] (p. 79)". While they do not specify and define racism as the public attitude of concern, they note that "citizens of Western countries (i.e., developed countries of Europe, North America, and Oceania) do not always regard refugees with compassion and focus on their safety. Instead, at times they approach refugees with intolerance, distrust, and contempt, partly because they believe there is a trade-off between the well-being of refugees and the well-being of established members of potential host countries [33] (p. 80)".

In their first literature review, the authors found that public attitudes toward refugees tend to construct refugees as threats. These constructs are: (1) threat to safety, such as the association of refugees with terrorism; (2) threats to the economy, such as bogus claimants who are only here for the money; (3) threats to culture, such as flawed beliefs about "how Muslims treat women;" and (4) threats to health, such as carriers of disease, particularly communicable disease. These perceptions of refugees as threats were found to be the strongest predictors of racism toward refugees [33] (p. 9).

Esses et al. also reviewed the literature on the factors influencing refugee mental health. This is an important area of inquiry since research demonstrates that refugees experience higher rates of mental health issues and mental illness than the population of their country of origin population, the host population, and other categories of newcomers. This may be due to the uniquely traumatic experiences refugees endure including trauma from violence, loss, and grief [33]. The factors that affect refugee mental health are divided into four areas: (i) refugee characteristics, (ii) pre-migration trauma, (iii) the resettlement process, and (iv) post-displacement factors. Many of the factors in the latter two categories, which occur in host countries, are examples of racism. These include time spent in detention, the asylum interview process, economic opportunities, host country language proficiency, and experiences of discrimination [33].

## 6. Racism as a Social Determinant of Health and Wellness of Newcomer Populations

To reiterate, the purpose of this paper is to demonstrate how racism affects the acculturation process for newcomers. We use John Berry's model of acculturation to demonstrate this. We begin by briefly describing the model and then demonstrating how racism disrupts newcomer acculturation.

John Berry's model of acculturation identifies four forms or paths of acculturation, that refugees often take. Among these pathways are: (1) Integration occurs when newcomers maintain their culture and values while adopting certain aspects of the host society's culture and values. This is often regarded as the most ideal form of acculturation [34]; because it allows people to maintain the core aspects their identity while adopting the values and practices of their host country that help them successfully navigate their new worlds (2) Assimilation occurs when newcomers reject their heritage and adopt most or all aspects of the host society's culture; (3) Separation occurs when newcomers maintain their heritage culture and values and reject that of the host society; and (4) Marginalization occurs when newcomers reject both their own and the host society's heritage [35].

We contend that racism affects newcomer populations at the site of acculturation, specifically on the paths they select or are obliged to pursue in response to their settlement experiences. These acculturation paths are, in part, coping strategies for refugees dealing with post-displacement stress and trauma, as shown in Paradies' (2006) five pathways between racism and health. According to Esses, Hamilton, and Gaucher (2017), "one of the major solutions to the refugee crisis must be refugee resettlement in new host countries" [33] (p. 78). This must involve more than simply allowing refugees to enter western countries. As the World Health Organization states, we must create an environment that promotes the mental health and wellness of incoming refugees. (WHO 2018) Integration is most conducive to this goal. When refugees experience racism, they are more likely to choose assimilation, separation, or marginalization as a coping strategy [35]. Evidence suggests that these coping strategies have detrimental effects on health and wellness [32].

In this section, we will compare each of the three scenarios of racism outlined previously to Dovidio, Gaertner, and Kawakami's (2010) definition of racism. We will also demonstrate how each scenario may lead to a less-than-ideal path of acculturation according to Berry's typology, as well as poor health and wellness based on Paradies' (2006) pathways. Each of these scenarios is provided as an example of possible outcomes, not as proof of the only conceivable pathways.

The establishment of Christmas as a statutory holiday is an example of cultural racism, as defined by Dovidio, Gaertner, and Kawakami (2010), whereby one racial group sets cultural standards for all. Assimilation may result from the stigmatization of cultural practices. Stigmatization often leads to shame and internalized racism, which is the adoption of racist views toward one's own race [13]. For example, consider the language of Canada's Zero Tolerance for Barbaric Cultural Practices Act of 2015. This Act was an amendment to the Immigration and Refugee Protection Act [33]. While the Act pertains to marriage practices, many Canadians have extended the notion of barbarism to other cultural practices followed by non-white newcomers. In this context of stigma, diverse cultural behaviours are frequently impossible to distinguish, rendering them all "barbaric" in the eyes of policy and mainstream discourse. This form of racism may lead a racialized minority newcomer to follow Berry's separation pathway, rejecting the cultural practices of their heritage to avoid shame. This scenario best exemplifies Paradies' second pathway, namely adverse cognitive and emotional experiences as a result of racist, stigmatizing language used in policy discourse [2].

Because it has unjust effects on racialized minorities, "carding" is an example of Dovidio, Gaertner, and Kawakami's (2010) concept of institutional racism [13]. The experience of carding can be traumatic, leading to feelings of fear and mistrust toward host societies [36–38]. Research has found that newcomers in the United States who are subjected to carding in host societies are more likely to develop feelings of hostility toward host cultures, as well as heightened identification with their heritage identity. Thus, they are more likely to adopt Berry's separation pathway of acculturation. In extreme cases, these practices, combined with a lack of supports and services for mental health, education, and employment during settlement, can lead to engagement in violence and extremist activities [33]. Outcomes associated with this scenario are most consistent with Paradies et al.'s (2006) first pathway, increased contact with police, and third pathway, injury as a result of racist violence [2].

The experiences of South Asian immigrant women in Australia illustrate Dovidio, Gaertner, and Kawakami's (2010) concept of individual racism [13]. We contend that, while race/ethnicity is the label that many organizations and literatures use, individual health is not determined by race itself. People are not at risk of poor health outcomes by virtue of their race, but rather because of unfair and unjust treatment based on their racialized identity. Patient-centered care, for example, is an empowering style of medicine that results in effective health outcomes for patients by involving patient values and preferences [39]. Racist perceptions of patient efficacy act as a barrier to patient-centered care, preventing many racialized refugees from making informed, supported decisions about their health. It

may also lead to Berry's marginalization path of acculturation, which includes a rejection of the host society's medical community as well as a rejection of (or lack of access to) one's heritage health practices. (Berry 1997) This form of acculturation is possibly the most alienating and dangerous, particularly in terms of health. This type of racism can create barriers to healthcare access and is most closely aligned with Paradies' 2006 fourth pathway to poor health, reduced participation in healthy behaviors [2], such as attending regular medical appointments.

## 7. Conclusions

While previous research has demonstrated that racism is a social determinant of health [1,2], little research has been conducted on how racism is a social determinant of health for newcomers during the acculturation process. The discussion in this paper has underscored that racism is a social determinant of health on a global scale. There is examples of this in Canada, the US, New Zealand, Australia, and European countries. Using Berry's typology, we illustrated (Figure 1) how racism functions as a social determinant of health by interfering with integration, the process of ideal acculturation. As a result of racism, refugees are more likely to be forced into assimilation, separation, or marginalization as a means of coping. These coping methods have a number of harmful physical and mental health consequences. According to the World Health Organization, it is the obligation of refugee-welcoming nations to promote the mental health and wellness of incoming refugees [40]. A fundamental means of supporting health and wellness is healthy acculturation (integration), which is heavily dependent on reducing the impact of cultural, institutional, and individual racism. This exploration demonstrates a need for future research in racial studies, examining the association between racism and acculturation for newcomers. Furthermore, our discussion necessitates a multi-sectoral approach that includes the entities discussed in this paper (immigration policy, healthcare service delivery, and policing), as well as other institutions and organizations that have an impact on population wellbeing. This contributes to the population public health goal of ensuring that all individuals, regardless of race, religion, or citizenship status, have access to optimal health.

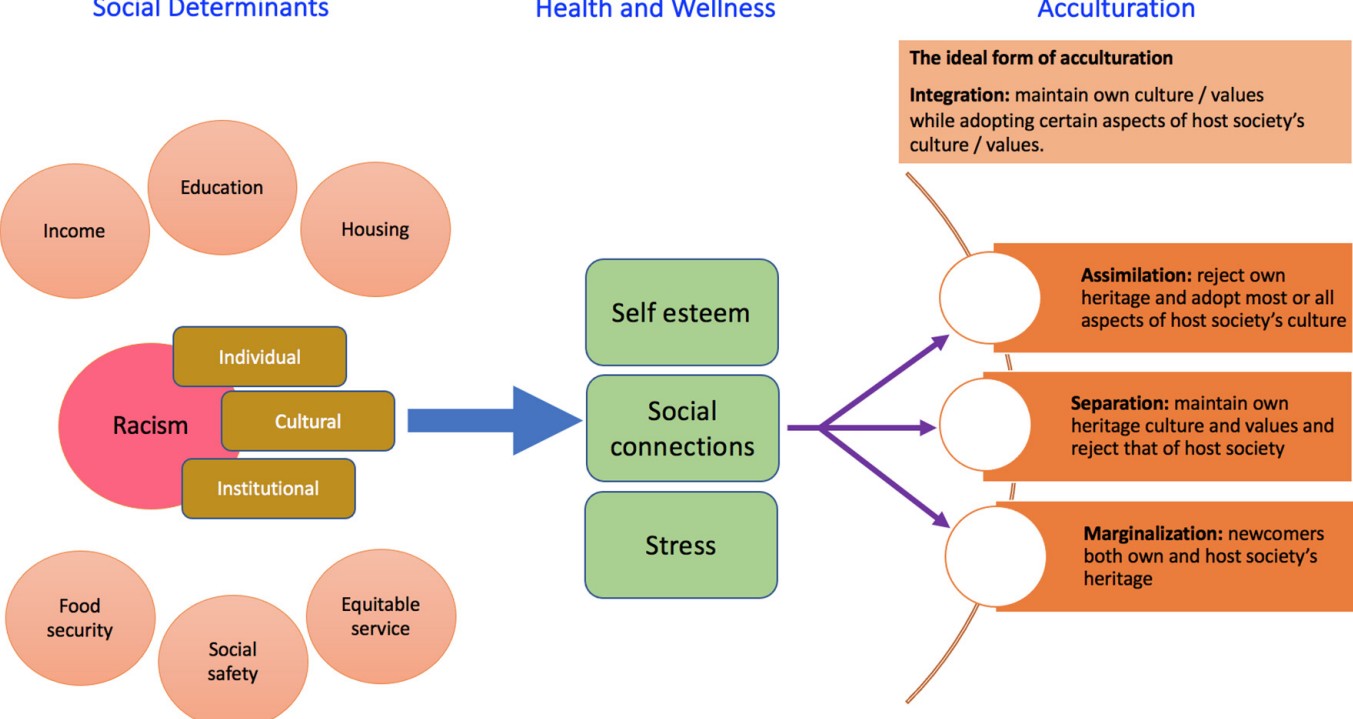

**Figure 1.** Racism as social determinant of health-wellness and acculturation of newcomers.

**Author Contributions:** J.N., E.O.P. and T.C.T. conceptualized this manuscript. J.N. drafted the manuscript. E.O.P. and T.C.T. provided intellectual inputs throughout the manuscript development process and critically reviewed the manuscript toward completion. All authors have read and agreed to the published version of the manuscript.

**Funding:** This research received no external funding.

**Institutional Review Board Statement:** Not applicable.

**Informed Consent Statement:** Not applicable.

**Conflicts of Interest:** The authors declare no conflict of interest.

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
