# Peer review of "Racism as a Social Determinant of Health for Newcomers towards Disrupting the Acculturation Process"

_societies, doi:10.3390/soc13010002_

Round 1

Reviewer 1 Report

This is an interesting paper on racism as a social determinant of health. Besides its merits, the paper need to be better anchored in current racial literature. This aspect is reflected in the short reference list of the paper.

Authors mentioned in the abstract that the paper is based on examples from newcommers in the US, Canada, New Zealand and European countries. If the US and Canadian cases are reflected in the literature review, there are very few touches in this paper on racial practices in European countries and New Zealand. For New Zealand it can be given the case of the maori people (see R Harris et al in journal Lancet, 2006, L Becares et al on a case of maori in Aotearoa), while for European countries there is the case of the Roma/Gypsy population. For issues of perpetuation of racial stigma there are the situation of the Roma communities in Hungary (see Méreiné Berki et al published in journal Cities, 2021, who showed that social mixing and desegregation in Szeged brought new patterns of stigma, also an article on internalised stigma in a disadvantaged neighbourhood published in Geographica Pannonica, 2020), or interesting issues of fragmented habitus and the everyday stigma in Czekia and Romania (see Ryan Powell et al., in  International Journal of Urban and Regional Research, 2022.) Migrant Roma people are also racialised (see the works of C Yıldız, N De Genova in Social Identities, 2018 or even the study of Jan Gill on Slovakian Roma in Britain). Even those Roma who want to flee marginalization are seen as others in a multicultural European borderland (Covaci and Jucu, 2021 in journal Identities).  

Besides the Roma people there are the case of black communities in western Europe who are also racialised, see Bhopal K. on racialised issues in rural England published in journal Gender and Education, 2014 and Joseph-Salisbury R in Journal of Negro Education, 2017, on how black people are racialised in the UK secondary education system.

Sometimes racial issues are used as mobilizing tools by the far right members (see Powell et al, 2018 in the case of wealthy Roma in journal IJURR  and OBrien et al 2019 on a case of mobilizing tool for expulsion of blemished people from the Traian square in a multicultural city). On the other hand, a particular case is that of internal migrants who are stigmatized due to their more black skin by natives in specific regions(see O’Brien, T. et al, 2022 in journal Identities).

Such examples as those presented above could be shortly mentioned both in the literature review and some of them even in the discussion of the paper.

Second, conclusions are a little bit short. It lacks the international implications of this study or how the novelty of this study pushes forwards what we know in racial studies.

Author Response

  • This is an interesting paper on racism as a social determinant of health. Besides its merits, the paper need to be better anchored in current racial literature. This aspect is reflected in the short reference list of the paper.

Thank you for your review and suggestions below. We have found your review comments to be very valuable in improving the international applicability of our manuscript and have made efforts to include your suggestions below.

  • Authors mentioned in the abstract that the paper is based on examples from newcommers in the US, Canada, New Zealand and European countries. If the US and Canadian cases are reflected in the literature review, there are very few touches in this paper on racial practices in European countries and New Zealand. For New Zealand it can be given the case of the maori people (see R Harriset al in journal Lancet, 2006, L Becares et al on a case of maori in Aotearoa), while for European countries there is the case of the Roma/Gypsy population. For issues of perpetuation of racial stigma there are the situation of the Roma communities in Hungary (see Méreiné Berki et al published in journal Cities, 2021, who showed that social mixing and desegregation in Szeged brought new patterns of stigma, also an article on internalised stigma in a disadvantaged neighbourhood published in Geographica Pannonica, 2020), or interesting issues of fragmented habitus and the everyday stigma in Czekia and Romania (see Ryan Powell et al., in  International Journal of Urban and Regional Research, 2022.) Migrant Roma people are also racialised (see the works of C Yıldız, N De Genova in Social Identities, 2018 or even the study of Jan Gill on Slovakian Roma in Britain). Even those Roma who want to flee marginalization are seen as others in a multicultural European borderland (Covaci and Jucu, 2021 in journal Identities).  

We have added the following content, per your suggestion, to include a New Zealand example similar to our Australian example of individual racism. (lines 121-124):

“A similar example can be found in the context of New Zealand. Harris, Cormack, and Stanley (2019) found that racism by health professionals, which often includes race based stereotyping and assumptions of health behaviours, leads to higher rates of unmet needs among racialized communities in New Zealand.”

We have also added content to demonstrate that this example extends beyond Canada, and has international implications (lines 82-84):

“One example of this is the declaration of Christmas day as a statutory holiday in Canada, the US, Australia, New Zealand, and European countries, with almost universal time off work and school to observe the tradition. While Christmas is celebrated by many racial groups in these countries…”

  • Besides the Roma people there are the case of black communities in western Europe who are also racialised, see Bhopal K.on racialised issues in rural England published in journal Gender and Education, 2014 and Joseph-Salisbury R in Journal of Negro Education, 2017, on how black people are racialised in the UK secondary education system.

We have added the following content, per your suggestion, to demonstrate institutional racism in the UK (lines 104-109):

“Institutional racism also occurs in school systems in the sense that curriculum design focuses disproportionately on ethnocentric perspectives of history, leaving out the voices and experiences of racialized people. This leads to students feeling like they are unseen and unheard within the school system, and often “home education” more valuable than that found in the school system. This disengagement and disenfranchisement can have serious impacts on school satisfaction and school performance. (Joseph-Salisbury 2017) 

  • Sometimes racial issues are used as mobilizing tools by the far right members (see Powell et al, 2018 in the case of wealthy Roma in journal IJURR  and O’Brien et al 2019 on a case of mobilizing tool for expulsion of blemished people from the Traian square in a multicultural city). On the other hand, a particular case is that of internal migrants who are stigmatized due to their more black skin by natives in specific regions(see O’Brien, T. et al, 2022 in journal Identities).

Thank you for your feedback and suggestions for improving the international focus of this paper. Please see above for added content, per your suggestions.

  • Such examples as those presented above could be shortly mentioned both in the literature review and some of them even in the discussion of the paper.

Thank you for your valuable suggestions. We have added this content to the paper.

  • Second, conclusions are a little bit short. It lacks the international implications of this study or how the novelty of this study pushes forwards what we know in racial studies.

Thank you for your feedback. We have added content to succinctly demonstrate the international implications and value of this paper to racial studies (lines 307-311; lines 320-321):

“While previous research has demonstrated that racism acts as a social determinant of health (Harrell et al. 2011; Paradies 2006), little research has been conducted on how racism is a social determinant of health during the acculturation process for newcomers. The discussion in this paper has demonstrated that racism is a social determinant of health on a global scale. Examples of this can be found in Canada, the US, New Zealand, Australia, and European countries.”  

“This exploration demonstrates a need for future research in racial studies, examining the association between racism and acculturation for newcomers.”

Reviewer 2 Report

The paper presented is suggestive. In fact, its purpose is extremely interesting and necessary. In this sense, I would like to congratulate the authors. However, I must point out that theoretical works need a lot of documentary support and debate with previous ideas. I think it is necessary that the authors make a greater effort to restructure the work and avoid some aspects that I will explain later.  The only purpose of indications I am going to make is to authors improving their proposal. I hope that they are not understood as a destruction of your researching, because I consider it more valuable and important. However, I humbly believe that they do not conform to the common standards.

It is somewhat curious the abstract states: "Research demonstrates that racism is a social determinant of health (SDOH), particularly for racialized minority newcomers residing in developed nations such as the United States, Canada, New Zealand, and European countries" and the first two sentences of the paper say: "Racism is a social determinant of health (SDOH) that is associated with poor physical and mental health outcomes. (Harrell et al. 2011; Paradies, 2006) Moreover, racism has a unique impact on racialized minority newcomers residing in developed nations such as the United States, Canada, New Zealand, and countries in Europe". The authors should be aware that they seem to be indicating that they are going to investigate something that has already been done. I suggest they change the statement in the abstract and put it in past this sentence or indicate that it is research conducted by others.

I suggest that the meaning of "racialized minority" or "racialized populations" be explained. The way this concept is used (without explanation) it seems to have a value dimension. In other words, racialised populations are populations that are discriminated against in one way or another. Nevertheless, the authors are using race concept in their paper. This leads us to think that the term race is going to be used from a biosocial perspective (biological subspecies), so the term does not seem negative per se. However, using the qualifier "racialised" gives the opposite impression. It would therefore be useful to clarify these aspects.

I think it would be important for the authors to take a position, in relation to the work of Hunt and Megyesi (2008), in order to defend the subsequent use of the concept of race in their work.

Hunt, L. M., & Megyesi, M. S. (2008). The ambiguous meanings of the racial/ethnic categories routinely used in human genetics research. Social science & medicine (1982), 66(2), 349-361. https://doi.org/10.1016/j.socscimed.2007.08.034

The authors set out "four key literature reviews that we have identified as critical to understanding the relationship between racism as a SDOH". In the first two of these, the information provided is not very clear. It would be necessary to introduce more quantitative information to support the objectives of the article.

On the other hand, the authors state the following: "we will compare each of the three scenarios of racism outlined previously to Dovidio, Gaertner, and Kawakami's (2010) definition of racism. Subsequently, we will demonstrate how they may lead to a non-ideal path of acculturation based on Berry's typology, as well as poor health and wellness based on Paradies (2006) pathways". The problem is such demonstration is unclear. In this sense, I consider that all conceptual research allows understanding, deepening, etc. but it has difficulties in demonstrating something. I suggest remake these ideas.

On the other hand, the analysis of the scenarios needs more documentary comparison in order to be more scientifically valid. The way in which paper is presented is a bit superficial. For example, the statement "The establishment of Christmas as a statutory holiday is an example of cultural racism, whereby one racial group sets cultural values for all" needs further development, as it could give the impression that any cultural process could become racist. In this sense, many classical sociological theories (Habermas, Luhmann, Giddens, and so on) have shown that society establishes social norms, I wonder if these norms could also be considered racist. Another example: "The experience of carding can be traumatic, leading to feelings of fear and mistrust toward host societies". This statement, as it stands, seems to relate to opinion and not so much to science. If something is traumatic, it is necessary to prove that it is traumatic. The authors do not do so.

In short, the authors would have to rework their paper and expand it substantially, using more literature and avoiding phrases that might sound like opinion. 

Author Response

  • The paper presented is suggestive. In fact, its purpose is extremely interesting and necessary. In this sense, I would like to congratulate the authors. However, I must point out that theoretical works need a lot of documentary support and debate with previous ideas. I think it is necessary that the authors make a greater effort to restructure the work and avoid some aspects that I will explain later.  The only purpose of indications I am going to make is to authors improving their proposal. I hope that they are not understood as a destruction of your researching, because I consider it more valuable and important. However, I humbly believe that they do not conform to the common standards.

Thank you for your review and suggestions below. We have found your review comments to be very valuable in improving our manuscript and have made efforts to include your suggestions below.

  • It is somewhat curious the abstract states: "Research demonstrates that racism is a social determinant of health (SDOH), particularly for racialized minority newcomers residing in developed nations such as the United States, Canada, New Zealand, and European countries" and the first two sentences of the paper say: "Racism is a social determinant of health (SDOH) that is associated with poor physical and mental health outcomes. (Harrell et al. 2011; Paradies, 2006) Moreover, racism has a unique impact on racialized minority newcomers residing in developed nations such as the United States, Canada, New Zealand, and countries in Europe". The authors should be aware that they seem to be indicating that they are going to investigate something that has already been done. I suggest they change the statement in the abstract and put it in past this sentence or indicate that it is research conducted by others.

Thank you for your feedback. We have changed our language to demonstrate that this is previous research (lines 4; lines 22):  

“Previous Research has demonstrated that racism is a social determinant of health (SDOH), particularly for racialized minority newcomers residing in developed nations such as the United States, Canada, New Zealand, and European countries.”

“Previous research has demonstrated that racism is a social determinant of health (SDOH) that is associated with poor physical and mental health outcomes.(Harrell et al. 2011; Paradies, 2006)”

  • I suggest that the meaning of "racialized minority" or "racialized populations" be explained. The way this concept is used (without explanation) it seems to have a value dimension. In other words, racialised populations are populations that are discriminated against in one way or another. Nevertheless, the authors are using race concept in their paper. This leads us to think that the term race is going to be used from a biosocial perspective (biological subspecies), so the term does not seem negative per se. However, using the qualifier "racialised" gives the opposite impression. It would therefore be useful to clarify these aspects.

Thank you for this feedback. We feel that this suggestion greatly improves our discussion and have included a new section to define/describe racialization and explain our use of the term (lines 128-146)”

“Racialization 

In this paper, we refer to the concept of racialization as opposed to race. We do this to signify that the health outcomes we discuss are not the product of one’s, biology, i.e. their racial or ethnic make up, but of the experience they have because of the way they are perceived and treated in social settings. “Racialization is the process of manufacturing and utilizing the notion of race in any capacity” (Dalal, 2002, p. 27). It is the complex social and cultural process through which individuals and groups are assigned a particular "race" and socially stratified on the basis of that race. Racialization has been and continues to involve a unfair and unequal social stratification, resulting in social and health inequities. As such, race itself is a social construct as opposed to an essential part of an individual or group. The use of race as a variable in studies of human beings has been considered questionable, (Braun, 2004) and even racist (Reitmanova, Gustafson, Ahmed, 2015; Esses, Veenvliet, Hodson, Mithic, 2008) Hunt and Megyesi (2008), for instance, conducted interviews with human genetic scientists who used race as a variable in their research. They found that the basis on which the researchers categorized individuals by race were nebulous and incoherent, and that despite claims to scientific neutrality, we live in a racialized society, which is to say that race is socially constructed. The authors conclude that “persisting in constructing scientific arguments based on highly ambiguous variables that are clearly laden with dubious social meanings, is of deep concern.”(Hunt and Megyesi, 2008 p11) This paper contributes the body of literature exploring health inequities based on racialization as opposed to health outcomes based on race, to elucidate that such health outcomes are unfair, avoidable, and socially produced, as opposed to an essential part of an individual’s biology. (Reitmanova, Gustafson, Ahmed, 2015; Esses, Veenvliet, Hodson, Mithic, 2008)”

  • I think it would be important for the authors to take a position, in relation to the work of Hunt and Megyesi (2008), in order to defend the subsequent use of the concept of race in their work.

Thank you for this suggestion. We have explored the work of Hunt and taken a position in relation to the use of “racialization” vs “race” it into our paper. Our additional section is above.

  • Hunt, L. M., & Megyesi, M. S. (2008). The ambiguous meanings of the racial/ethnic categories routinely used in human genetics research. Social science & medicine (1982), 66(2), 349-361. https://doi.org/10.1016/j.socscimed.2007.08.034

Thank you for providing this reference. We have explored the work of Hunt and incorporated it into our paper.

  • The authors set out "four key literature reviews that we have identified as critical to understanding the relationship between racism as a SDOH". In the first two of these, the information provided is not very clear. It would be necessary to introduce more quantitative information to support the objectives of the article. On the other hand, the authors state the following: "we will compare each of the three scenarios of racism outlined previously to Dovidio, Gaertner, and Kawakami's (2010) definition of racism. Subsequently, we will demonstrate how they may lead to a non-ideal path of acculturation based on Berry's typology, as well as poor health and wellness based on Paradies (2006) pathways". The problem is such demonstration is unclear. In this sense, I consider that all conceptual research allows understanding, deepening, etc. but it has difficulties in demonstrating something. I suggest remake these ideas.

Thank you for this feedback. We have included content and citations in the paper throughout, to make our demonstration of these relationships clearer. These changes can be found throughout lines 254 to 303.

  • On the other hand, the analysis of the scenarios needs more documentary comparison in order to be more scientifically valid. The way in which paper is presented is a bit superficial. For example, the statement "The establishment of Christmas as a statutory holiday is an example of cultural racism, whereby one racial group sets cultural values for all" needs further development, as it could give the impression that any cultural process could become racist. In this sense, many classical sociological theories (Habermas, Luhmann, Giddens, and so on) have shown that society establishes social norms, I wonder if these norms could also be considered racist. Another example: "The experience of carding can be traumatic, leading to feelings of fear and mistrust toward host societies". This statement, as it stands, seems to relate to opinion and not so much to science. If something is traumatic, it is necessary to prove that it is traumatic. The authors do not do so.

Thank you for this valuable feedback. We have explored the work of Jyoshi 2020, and revised/added to our content based on our exploration to demonstrate that these statements are not opinion (lines 82-91). We have also explored the work of Tobias and Joseph, 2020, and have incorporated referencing to demonstrate that these statements are not opinion (lines 277-278).

“One example of this is the declaration of Christmas day as a statutory holiday in Canada, the US, Australia, New Zealand, and European countries, with almost universal time off work and school to observe the tradition. While Christmas is celebrated by many racial groups in these countries, it is historically a “white” holiday with European roots; the adoption of Christmas by racialized minorities is directly linked to colonialism. Moreover, the aforementioned countries identify as multi-cultural mosaic societies, with many racialized minority communities celebrating religious and cultural holidays other than Christmas, which are not generally recognized as a statutory holiday. Recognizing Christmas as a statutory holiday whilst not recognizing other, commonly observed holidays as statutory holidays serves to establish Christmas a norm, while “othering” holidays such as Eid al Fitr, Diwali, and Kwanzaa. (Jyoshi 2020)

“The experience of carding can be traumatic, leading to feelings of fear and mistrust toward host societies.(Tobias and Joseph, 2020)”

  • In short, the authors would have to rework their paper and expand it substantially, using more literature and avoiding phrases that might sound like opinion. 

Thank you for your thoughtful and valuable feedback. We feel that your input greatly improves our manuscript and we have made revisions per your suggestions.

Round 2

Reviewer 1 Report

Authors have improved their paper, but the major problem of this concept paper is that it still has very few cited references. There are only 23 cited references at the moment in the reference list - usually a concept paper should have minimum 40-50 references. For instance, I proposed the authors lots of studies on racialization and health aspects related to a racialized ethnic group in Eastern Europe (ie. the Roma people)  and I do not find any of those sources in the revised version (see aspects connected to the perpetuation of racialized stigma in Szeged, Hungary, see aspects of racialized urban encounters - doi -10.1111/1468-2427.13053 or even see otherness in rural spaces  - doi 10.1080/1070289X.2021.1920774 and so on). Besides these sources, I think the authors need to be engaged also with other more studies related to race and health issues in order to better position their concept study in the broader international racial studies. 

Author Response

Thank you for your second review and suggestions. We have taken the following steps in response to your suggestions:

  • We have explored and incorporated more references to support our work throughout the manuscript
  • We have added a discussion, three of the five you’re your suggested sources on the racialization of Roma communities in Europe were included.
  • We have expanded our literature review

These additions have brought us to include 40 sources in our paper. All the changes can be found throughout the manuscript in blue italics.

Reviewer 2 Report

The paper has improved substantially. I believe that the authors could have gone more deeply into the object of study. This research is extremely interesting so I would have liked more discussion and contrast with other works. However, this does not detract from the quality of the work. Therefore, I believe that the changes made are sufficient for publication.

Author Response

Thank you for your review and suggestions, we have included several additions to the manuscript in order to deepen our and broaden our discussion. All the changes can be found throughout the manuscript in blue italics.